

# Probabilistic characteristics of narrow-band long wave run-up onshore
**Sergey Gurbatov[1] and Efim Pelinovsky[2,3]**
1) National Research University – Lobachevsky State University, Nizhny Novgorod, Russia
2) National Research University – Higher School of Economics, Moscow, Russia
3) Institute of Applied Physics, Nizhny Novgorod, Russia
**Abstract**
The run-up of random long wave ensemble (swell, storm surge and tsunami) on the constant-
slope beach is studied in the framework of the nonlinear shallow-water theory in the
approximation of non-breaking waves. If the incident wave approaches the shore from deepest
water, runup characteristics can be found in two stages: at the first stage, linear equations are
solved and the wave characteristics at the fixed (undisturbed) shoreline are found, and, at the
second stage, the nonlinear dynamics of the moving shoreline is studied by means of the
Riemann (nonlinear) transformation of linear solutions. In the paper, detail results are obtained
for quasi-harmonic (narrow-band) waves with random amplitude and phase. It is shown that the
probabilistic characteristics of the runup extremes can be found from the linear theory, while the
same ones of the moving shoreline - from the nonlinear theory. The role of wave breaking due to
large-amplitude outliers is discussed, so that it becomes necessary to consider wave ensembles
with non-Gaussian statistics within the framework of the analytical theory of non-breaking
waves. The basic formulas for calculating the probabilistic characteristics of the moving
shoreline and its velocity through the incident wave characteristics are given. They can be used
for estimates of the flooding zone characteristics in marine natural hazards.

**Keywords:** tsunami, storm surge, long wave runup, Carrier-Greenspan transform, statistical
characteristics

## 1. Introduction
The flooded area size, the water flow depth and its speed on the coast, the coastal topography
characteristics and the features of the coastal zone development determine the consequences of
marine natural disasters on the coast. The catastrophic events of recent years are well known,
when tsunami waves and storm surges caused significant damage on the coast and people's
death. It is worth saying that only in 2018 two catastrophic tsunamis occurred in Indonesia,
leading to the death of several thousand people (on Sulawesi Island in September and in the
Sunda Strait in December). The calculations of the coast flooding in tsunamis and storm surges
is mainly carried out within the framework of nonlinear shallow-water equations, taking into
account the variable roughness coefficient for various areas of the coastal zone (Kaiser et al,



2011; Choi et al, 2012). The characteristics of the coastal destruction is determined either by
using fragility curves (Macabuag et al, 2016; Park et al, 2017) or by using a direct calculation of
the tsunami forces (Qi et al, 2014; Ozer et al, 2015a, b; Kian et al, 2016; Xiong et al., 2019).
The computation accuracy was tested on a series of benchmarks, including the idealized
problem of the wave run-up onto the impenetrable slope of a constant gradient without friction
(Synolakis et al, 2008). The nonlinear shallow water equations for the bottom geometry of this
kind are linearizedby using the hodograph (Legendre) transformations. This step makes it
possible to obtain a number of exact solutions describing the run-up on the coast. This approach,
first suggested in (Carrier and Greenspan, 1958), was later on used to analyze the run-up of
single and periodic waves of various shapes (Synolakis, 1987; Pelinovsky and Mazova, 1992;
Tinti and Toniti, 2005; Madsen and Fuhrman, 2008; Madsen and Schaffer, 2010; Antuano and
Brocchini, 2008, 2010; Didenkulova, 2009; Dobrokhotov et al, 2015; Aydin and Kanoglu, 2017).
Moreover, such approach made it possible to determine the conditions for the wave breaking.
The latter means the presence of steep fronts (gradient catastrophe) within the hyperbolic
shallow water equation framework. The Carrier-Greenspan transformation was further
generalized for the case of waves in an inclined channel of an arbitrary variable cross section
(Rybkin et al, 2013; Pedersen, 2016; Shimozone, 2016; Anderson et al, 2017; Raz et al, 2018). In
a number of practical cases, its use proves to be more efficient than the direct numerical
computation within the 2D shallow water equation framework (Harris et al, 2015, 2016).
Due to bathymetry variability and shoreline complexity, diffraction and scattering effects
lead to an irregular shape of the waves approaching the coast. Moreover, very often not the
leading wave turns out to be the maximum one. Such typical tsunami wave records on tide-
gauges are well known and are not shown here. It is applied even more to swell waves, which in
some cases approach the coast without breaking (Huntley et al, 1977; Hughes et al, 2010). As a
result, statistical wave theory can be applied to such records and with their help, nonlinear
shallow water equations in the random function class can be solved. This approach was used to
describe the statistical moments of the long wave run-up characteristics in (Didenkulova et al,
2008, 2010, 2011). Special laboratory experiments were also conducted on irregular wave run-up
on a flat slope, the results of which are not very well described by theoretical dependencies
(Denissenko et al, 2011, 2013). As for field data, we are acquainted with two papers: (Huntley et
al, 1977; Hughes et al, 2010), where the statistical characteristics of the moving shoreline on two
Canadian and one Australian beaches were calculated. They confirmed the fact that the wave
process on the coast is not Gaussian. In our opinion, the main problem in the theoretical model of
describing the irregular wave run-upon the shore is associated with the use of two hypotheses: 1)





the small amplitude wave field (in the linear problem) is Gaussian; 2) waves run-up on the shore
without breaking. It is obvious, however, that in the nonlinear wave field some broken waves can
always be present. They affect the distribution function tails and, thus, the statistical moments of
the run-up characteristics as well.
The connection of the run-up parameters at the nonlinear stage with the linear field at a
fixed point is described either in a parametric form or implicitly in a nonlinear equation
(Didenkulova et al. 2010). This does not allow using the standard methods of random processes.
At the same time, it is known that this implicit equation is equivalent to a partial first-order
differential equation (PDE), that is, to the simple or Riemann wave equation (Rudenko and
Soluyan, 1977). In statistical problems, this equation arises in nonlinear acoustics. This equation
or its generalization, the nonlinear diffusion equation called the Burgers equation (Burgers at al,
1974) is the model equation in the hydrodynamic turbulence theory (Frisch, 1995). It should be
noted that for the one-dimensional Burgers turbulence, as well as its three-dimensional version,
used for the model description of the large-scale Universe structure (Gurbatov et al, 2012). It is
possible to give an almost comprehensive statistical description for certain initial conditions
(Gurbatov et al, 1991, 1997, 2011; Gurbatov and Saichev, 1993; Molchanov et al, 1995; Frisch,
1995; Woyczynski, 1998; Frisch and Bec, 2001; Bec and Khanin, 2007). In particular, single-
point and two-point probability distributions of the velocity field and even $N$-point probability
distributions and, accordingly, multi-point moment functions were found. This partially allows
using a mathematical approach developed in statistical nonlinear acoustics. An experimental
study of the nonlinear evolution of random quasi-monochromatic waves and the probability
distributions and spectra analysis have been carried out in acoustics more than once. They
confirmed theoretical conclusions; see, for example (Gurbatov et al, 2018, 2019).
This paper is devoted to the analytical study of the probabilistic characteristics of the long
narrow-band wave run-upon the coast. Section 2 gives the basic equations of nonlinear shallow
water theory and the Carrier-Greenspan transformation, with the latter making it possible to
linearize the nonlinear equations. Section 3 describes the moving shoreline dynamics when the
deterministic sine wave climbs on the slope. The probability characteristics of the deformed sine
oscillations of the moving shoreline with a random phase are described in Sect. 4. Section 5
contains the probabilistic characteristics on the vertical displacement of the moving shoreline if
the incident narrow-band wave has a random amplitude and phase. The discussion of the wave
breaking effects and their influence on the distribution of the run-up characteristics is given in
Sec. 6. The results obtained are summarized in Sect. 7.

## 2. Basic equations and transformations

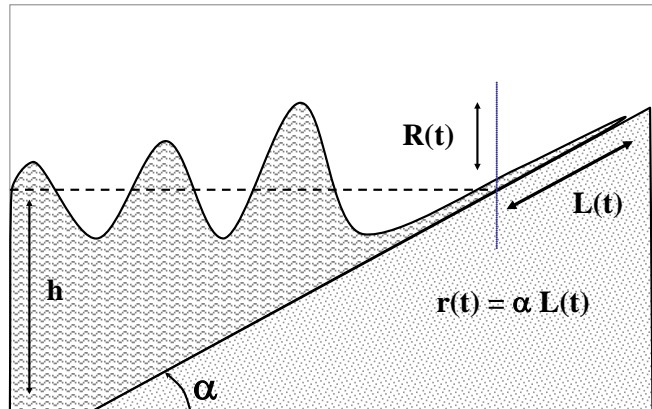

Fig. 1. The problem geometry

Here we will consider the classical formulation of the problem of a long wave run-upon the constant-gradient slope in an ideal fluid (Fig. 1). The wave is one-dimensional and propagates along the *x*-axis directed onshore. The basin depth is a linear depth function: $h(x) = -\alpha x$, where $\alpha$ is the inclination angle tangent and point *x = 0* corresponds to a fixed unperturbed water shoreline. *L(t)* and *r(t)* describe horizontal and vertical displacement of the moving shoreline, and *R(t)* is the water level oscillations at *x = 0*. The bottom and the shore are assumed impenetrable. The long wave dynamics is described by nonlinear shallow water equations:

$$\frac{\partial u}{\partial t} + u \frac{\partial u}{\partial x} + g \frac{\partial \eta}{\partial x} = 0, \qquad (2.1)$$

$$\frac{\partial \eta}{\partial t} + \frac{\partial}{\partial x}\big[(-\alpha x + \eta)u\big] = 0. \qquad (2.2)$$

Here, $\eta(x,t)$ is the free surface elevation above the undisturbed water level, and $u(x,t)$ is the depth-averaged flow velocity (within the shallow water theory, the flow velocity is the same on all horizons), and *g* is the gravity acceleration. Obviously, after introducing total depth

$$H(x,t) = -\alpha x + \eta(x,t), \qquad (2.3)$$

equations (2.1) and (2.2) are a hyperbolic system with constant coefficients. This fact makes it possible to transform the system into a linear equation one by using a hodograph (Legendre) transformation, which was done in the pioneering work (Carrier and Greenspan, 1958). As a

result, the wave field is described by a linear wave equation in the 'cylindrical' coordinate
system

$$\frac{\partial^2 \Phi}{\partial \lambda^2} - \frac{\partial^2 \Phi}{\partial \sigma^2} - \frac{1}{\sigma}\frac{\partial \Phi}{\partial \sigma} = 0, \qquad (2.4)$$

and all variables are expressed in terms of an auxiliary wave function $\Phi(\sigma, \lambda)$ using explicit
formulas

$$\eta = \frac{1}{2g}\left(\frac{\partial \Phi}{\partial \lambda} - u^2\right), \qquad (2.5)$$

$$u = \frac{1}{\sigma}\frac{\partial \Phi}{\partial \sigma}, \qquad (2.6)$$

$$x = \frac{1}{2\alpha g}\left(\frac{\partial \Phi}{\partial \lambda} - u^2 - \frac{\sigma^2}{2}\right), \qquad (2.7)$$

$$t = \frac{1}{\alpha g}(\lambda - u). \qquad (2.8)$$

It should be noted that the variable $\sigma$ is proportional to the total water depth.

$$\sigma = 2\sqrt{gH} = 2\sqrt{g(-\alpha x + \eta)}, \qquad (2.9)$$

so, the wave equation (2.4) is solved on the semi-axis $\sigma \geq 0$, and this coordinate plays the radius
role in the cylindrical coordinate system. We would like to emphasize that the point $\sigma = 0$
corresponds to a moving shoreline, and therefore, the original problem, solved in the area with a
unknown boundary, is reduced to a fixed area problem.

It is important to note that the hodograph transformation is valid if the Jacobian
transformation is non-zero

$$J = \frac{\partial(x,t)}{\partial(\sigma,\lambda)} \neq 0. \qquad (2.10)$$

It is the case when a gradient catastrophe, identified in the framework of the shallow-water
theory with the wave breaking, does not occur. The necessary condition for the wave breaking
absence is the boundedness and smoothness of all solutions; this question will be discussed
further on.

We will assume that the wave approaches the coast from the area far from the shoreline (
$x \to -\infty$ ), where the wave is linear. Then it is obvious that the function $\Phi(\sigma, \lambda)$ can be
completely found from the linear theory. The difficulty in finding the wave field in the near-





shoreline area is due to the implicit transformation of the coordinates *(x,t)* to $(\sigma, \lambda)$. However,
for the most interesting point of the moving shoreline $\sigma = 0$ (its dynamics determines the size of
the flooded area on the coast) all the formulas become explicit. In particular, from (2.5) and (2.6)
follows

$$r(t) = R\left[t + \frac{u(t)}{\alpha g}\right] - \frac{u(t)^2}{2g} \ , \tag{2.11}$$

$$u(t) = U\left[t + \frac{u(t)}{\alpha g}\right], \tag{2.12}$$

where *r(t)* and *u(t)* are the vertical displacement of the moving shoreline and its speed, and the
functions *R(t)* and *U(t)* determine the field characteristics at the fixed point *(x = 0)* from the
linear theory

$$R(t) = \frac{1}{2g}\frac{\partial \Phi(\sigma = 0, \lambda)}{\partial \lambda}\bigg|_{\lambda = \alpha g t} \ , \qquad U(t) = \frac{1}{\sigma}\frac{\partial \Phi(\sigma, \lambda)}{\partial \sigma}\bigg|_{\sigma = 0, \lambda = \alpha g t} \ . \tag{2.13}$$

Then we add the obvious kinematic relations for the vertical displacement and velocity of the last
sea point along the slope.

$$u(t) = \frac{1}{\alpha}\frac{dr(t)}{dt} \ , \qquad U(t) = \frac{1}{\alpha}\frac{dR(t)}{dt}. \tag{2.14}$$

Let us note that formula (2.12) is identical to the so-called Riemann wave or a simple

wave in a nonlinear non-dispersive medium (in particular, in nonlinear acoustics), if we consider
the parameter $1/\alpha g$ to be a 'coordinate'; see, for example, (Rudenko and Soluyan, 1977,
Gurbatov et al, 1991, 2011). Moreover, formula (2.13) describes the integral over the Riemann
wave. This analogy proves to be very useful when transferring the already known results in the
wave nonlinear theory to the run-up characteristics described by the ODE.

Detailed calculations of the long wave run-up on the coast were carried out repeatedly;

see, for example (Carrier and Greenspan, 1958; Synolakis, 1987; Pelinovsky and Mazova, 1992;
Tinti and Toniti, 2005; Madsen and Fuhrman, 2008; Madsen and Schaffer, 2010; Antuano and
Brocchini, 2008, 2010; Didenkulova, 2009; Dobrokhotov et al, 2015; Aydin and Kanoglu, 2017).

It is worth mentioning that the nonlinear time transformation in (2.11) and (2.12) leads to

the shoreline oscillation distortion in comparison with the linear theory predictions. So, for large
amplitudes the wave shape becomes multi-valued (broken). The first moment of the wave




breaking on the shoreline (the gradient catastrophe) is easily found from (2.12) by calculating the
first derivative of the moving shoreline velocity

$$\frac{du}{dt} = \frac{dU/dt}{1 - \dfrac{dU/dt}{\alpha g}}, \qquad (2.15)$$

from it follows the wave breaking condition

$$Br = \frac{\max(dU/dt)}{\alpha g} = \frac{\max(d^2 R/dt^2)}{\alpha^2 g} = 1, \qquad (2.16)$$

where we have introduced the breaking parameter – $Br$ to designate the left-hand side in (2.16),
which characterizes the nonlinear wave properties on the shoreline. The condition (2.16) can be
given a physical meaning, that the breaking occurs when the last sea particle acceleration (
$\alpha^{-1} d^2 R/dt^2$ ) exceeds the component of gravity acceleration along the shoreline ( $g\alpha$ ). As
shown in (Didenkulova, 2009), condition (2.16) coincides with (2.10) for Jacobian. It is
important to emphasize that the breaking condition is unequivocally found through solving the
linear problem of the wave run-up on the shore. It is determined only by the particle acceleration
value on the shoreline; but it is not determined separately by the shoreline displacement or its
velocity.

A similar Carrier – Greenspan transformation is obtained for waves in narrow inclined

channels, fjords, and bays (Rybkin et al, 2013; Pedersen, 2016; Anderson et al, 2017; Raz et al,
2018); only the wave equation (2.4) and relations (2.5) - (2.8) change. However, the moving
shoreline dynamics is still described by equations (2.11) and (2.12), valid for arbitrary cross-
section channels.

### 3. The moving shoreline dynamics at an initially monochromatic wave run-up

The monochromatic wave run-up on a flat slope by using the Carrier – Greenspan

transformation has been studied in a number of papers cited above. Let us reproduce here the
main features of the moving shoreline dynamics necessary for us to draw a statistical description
further on. Mathematically, the monochromatic wave run-up is described by an elementary
solution of equation (2.4)

$$\Phi(\sigma, \lambda) = Q J_0(l\sigma) \cos(l\lambda), \qquad (3.1)$$




where $Q$ and $l$ are arbitrary constants, and $J_0$ is the zero-order Bessel function. Far from the
shoreline ($\sigma \to \infty$) the Bessel function decreases, so the wave function $\Phi$ becomes small. In this
case, in (2.5) - (2.8) one can use approximate expressions (the 'linear' Carrier – Greenspan
transformation)
$$\eta = \frac{1}{2g}\frac{\partial \Phi}{\partial \lambda}, \qquad u = \frac{1}{\sigma}\frac{\partial \Phi}{\partial \sigma}, \qquad x = -\frac{\sigma^2}{4\alpha g}, \qquad t = \frac{\lambda}{\alpha g}, \tag{3.2}$$

and using the asymptotic representation for the Bessel function, reduce (3.1) to the expression
for the water surface displacement
$$\eta(x,t) = a(x)\left\{ \sin\left[ \omega\left( t - \int \frac{dx}{\sqrt{gh(x)}} \right) \right] - \frac{\pi}{4} \right\} + \sin\left[ \omega\left( t + \int \frac{dx}{\sqrt{gh(x)}} \right) + \frac{\pi}{4} \right], \tag{3.3}$$

where
$$a(x) = \frac{Q}{2g}\sqrt{\frac{l}{\pi\sqrt{gh(x)}}}, \qquad \omega = gl\alpha . \tag{3.4}$$

The wave field away from the shoreline is a superposition of two waves of the same frequency
and a variable amplitude $a(x)$, which together form a standing wave. It immediately shows that
the wave amplitude varies with depth according to the Green law ($h^{-1/4}$), as it should be far from
the coast. The same asymptotic result follows from the exact solution of linear shallow water
equations.
$$\eta(x,t) = R_0 J_0\left( \sqrt{\frac{4\omega^2 |x|}{g\alpha}} \right) \sin(\omega t) , \tag{3.5}$$

where $R_0$ is the wave amplitude at the fixed shoreline ($x = 0$), identified with the maximum run-
up height in the linear theory. By connecting (3.4) and (3.5), we obtain the formula for the run-
up height obtained through the incident wave amplitude far from the coast
$$\frac{R_0}{a(x)} = \sqrt{\frac{2\omega}{\alpha}\sqrt{\frac{h(x)}{g}}} . \tag{3.6}$$

Formula (3.6) allows working further with the run-up height $R_0$ instead of the wave amplitude far
from the coast $a(x)$, considering it to be given. Having determined $Q$ and $l$ through the incident
wave parameters, we can calculate the run-up characteristics in the nonlinear theory, considering
the limit of formula (3.1) with $\sigma \to 0$ and using the Carrier – Greenspan transformation formulas
(2.5) - (2.8). The moving shoreline movement is determined by the parametric dependence
$$t = \frac{\lambda}{\alpha g} - \frac{\omega R_0}{\alpha^2 g} \cos\left(\frac{\omega\lambda}{\alpha g}\right), \qquad (3.7)$$

$$r = R_0 \sin\left(\frac{\omega\lambda}{\alpha g}\right) - \frac{\omega^2 R_0^2}{2\alpha^2 g} \cos^2\left(\frac{\omega\lambda}{\alpha g}\right). \qquad (3.8)$$

It is convenient to introduce dimensionless variables
$$z = \frac{r}{R_0}, \qquad \tau = \omega t. \qquad \varphi = \frac{\omega\lambda}{\alpha g}, \qquad (3.9)$$

and calculate the breaking parameter
$$Br = \frac{\omega^2 R_0}{\alpha^2 g}, \qquad (3.10)$$

so the formulas (3.7) and (3.8) are finally rewritten in the form
$$\tau = \varphi - Br \cos(\varphi), \qquad (3.11)$$

$$z = \sin(\varphi) - \frac{Br}{2} \cos^2(\varphi), \qquad (3.12)$$

what is another record for the formulas (2.11) and (2.12), if we take
$$R(t) = R_0 \sin(\omega t), \qquad (3.13)$$

arising from (3.5) with $x = 0$. Let us note that the function $z(\tau, Br)$ is set in a parametric form,
but after expressing $\varphi$ from (3.12) and substituting it in (3.11), we can obtain the explicit
expression for the function $\tau(z; Br)$. In the paper, we will use both explicit and implicit
expressions of the functions describing the moving shoreline dynamics.

Fig. 2 shows the moving shoreline dynamics at different wave height values in terms of

the breaking parameter up to the limiting value ($Br = 1$). In the limit of small parameter values,
the oscillations are close to sinusoidal (it is almost a linear problem). Then, with the increasing
amplitude, the moving shoreline velocity gets a steep leading front, while at the moving
shoreline vertical displacement a peculiar feature is formed at the wave run-down stage. As it is
known, at the time of the Riemann wave breaking, a peculiarity like $u \sim t^{1/3}$ is formed
(Pelinovsky et al, 2013). Then, in the integral over the Riemann wave (at the moving shoreline
displacement), this peculiar feature will have the form $z \sim t^{4/3}$. Thus, with the wave amplitude
increase, the first breaking occurs at sea (at the run-down stage), and not on the coast. Then the
breaking zone expands and moves on to the coast, but at this stage, analytical solutions based on
the Carrier-Greenspan transformation become inapplicable.


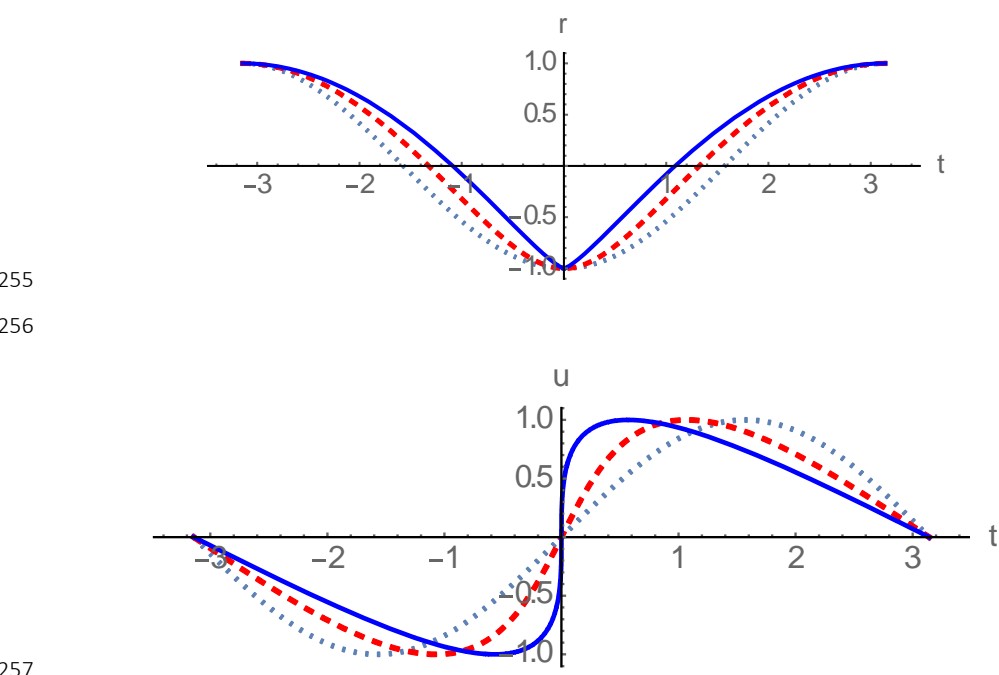





Fig. 2. The moving shoreline dynamics (top) and its velocity (below) for different breaking
parameter values *Br* (0 –the dotted line, 0.5 –the dashed line and 1 – the solid line).

**4. Probabilistic characteristics of the initially sine wave run-up with a random phase**
Let us now consider the probabilistic characteristics of the initially sine wave run-up with a
random phase on the shore, assuming it to be uniformly distributed over the interval $[0-2\pi]$.
These characteristics are found by using the geometric probability methods (Kendall and Stuart,
1969), so that for ergodic processes the probability density of the moving shoreline vertical
displacement coincides with the relative location time of the function $\xi(t)$ in the interval ($\xi$,
$\xi + d\xi$)

$$W(\xi) = \frac{1}{2\pi} \sum_{n=1}^{N} \left| \frac{dt_n}{d\xi} \right|,$$ (4.1)

where the summation takes place at all intersection level $\xi(t)$. For harmonic disturbance, it is
enough to restrict ourselves to considering the field on a half-period. So, for the moving
shoreline vertical displacement in dimensionless variables, the derivative $d\tau / dz$ of the




parametric curve (3.11) and (3.12) can be calculated through the ratio of the derivatives $d\tau/d\varphi$
и $dz/d\varphi$

$$W_z^{\sin}(z;Br) = \frac{1}{\pi} \frac{1 + Br \sin\varphi}{\cos\varphi + Br \cos\varphi \sin\varphi} = \frac{1}{\pi \cos\varphi} \quad , \tag{4.2}$$

we indicated here that the probability density depends on $Br$ as a parameter. Finding $\cos\varphi$ from
the formula (3.12) for the vertical displacement, we obtain the final expression for the
probability density

$$W_z^{\sin}(z;Br) = \frac{1}{\pi} \frac{1}{\sqrt{1 - \frac{1}{Br^2}\left[1 - \sqrt{1 + 2zBr + Br^2}\right]^2}} \quad , \tag{4.3}$$

which in the linear problem for a purely sinusoidal perturbation transforms into a well-known
expression for the probability distribution of a harmonic signal with a random phase (Kendall
and Stuart, 1969)

$$W_z^{\sin}(z;0) = \frac{1}{\pi} \frac{1}{\sqrt{1 - z^2}} \quad . \tag{4.4}$$

The probability distribution (4.3) for the three values of the parameter $Br$ is shown in Fig.
3. As you can see, the probability density becomes an asymmetric function with a greater
probability in the area of positive values corresponding to the wave run-up on the coast than at
the run-down stage. At the ends of the interval, the probability density is unlimited throughout
the entire range change of $Br$, since the shoreline oscillations near the maximum have a zero
derivative (the moving shoreline velocity in it becomes zero).
The obtained probability density function can be used to calculate the statistical moments
of the shoreline oscillations. Technically, however, it is easier to use the parametric equations
(3.11) and (3.12) and calculate all the moments.

$$M_n^z = \frac{1}{2\pi} \int_0^{2\pi} z^n(\tau)d\tau = \frac{1}{2\pi} \int_0^{2\pi} z^n(\varphi) \frac{d\tau}{d\varphi} d\varphi \quad . \tag{4.5}$$

So, the first moment

$$M_1^z = \frac{Br}{4} \tag{4.6}$$

determines the average water level rise on the coast when the waves approach the shore (set-up
phenomenon), which is commonly observed (Dean and Walton, 2009).

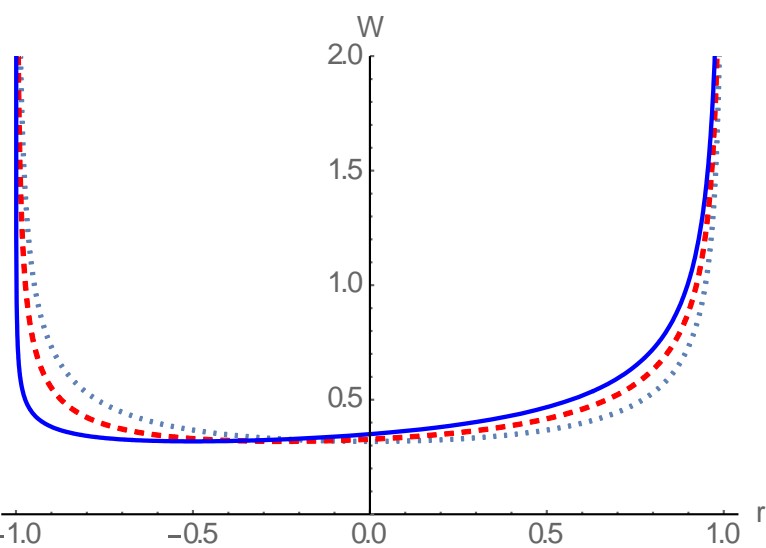


Fig. 3. The probability density of the moving shoreline vertical displacement for the initially sine
wave run-up at $Br = 0$ (the dotted line), 0.5 (the dashed line) and 1 (the solid line).


The second moment determines the dispersion

$$\delta^2 = \frac{1}{2\pi} \int_0^{2\pi} (z - M_1^z)^2 \, d\tau = \frac{1}{2} - \frac{3}{32} Br^2, \qquad (4.7)$$

characterizing the fluctuation range relative to the average value; it relatively weakly decreases
with the growth of the parameter $Br$ (less than 10% for non-breaking waves).

Finally, the total flooding time and its drainage time are easy to find from (3.11) and
(3.12), finding from the equation mentioned last, the value $\varphi$, at which $z = 0$, and substituting the
obtained values in (3.11)

$$T_{flood} = \pi - 2\arcsin\left[\frac{\sqrt{1+Br^2}-1}{Br}\right] + 2\sqrt{2}\sqrt{\sqrt{1+Br^2}-1},$$

(4.9)

$$T_{dry} = \pi + 2\arcsin\left[\frac{\sqrt{1+Br^2}-1}{Br}\right] - 2\sqrt{2}\sqrt{\sqrt{1+Br^2}-1},$$

Both times change almost linearly with the increasing wave amplitude (parameter $Br$), see Fig. 4.

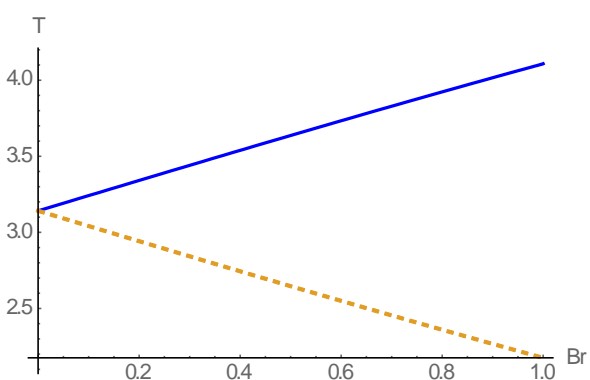

Fig. 4. The total flooding time (the solid curve) and the drainage time (the dashed curve) depending on the parameter *Br*.

It is worth noting that, in contrast to the vertical displacement, the moving shoreline velocity distribution [$u = (\omega R_0 / \alpha)v$], as it is easy to show, does not depend on the breaking parameter and probability density function is determined by the simple formula

$$W_v^{\sin}(v) = \frac{1}{\pi} \frac{1}{\sqrt{1-v^2}} . \qquad (4.10)$$

The distribution independence on the degree of nonlinearity is well known for the Riemann waves and is explained by the compensation of compression and rarefaction areas (Gurbatov et al, 1991, 2011).

## 5. Probabilistic characteristics of a narrow-band wave run-up with a random amplitude and phase

Let us consider the run-up of a quasi-harmonic wave with a random amplitude and phase on a flat slope. To do this, we will first rewrite formulas (4.3) and (4.10) for them to include the wave amplitude. It is convenient to enter the maximum height $R_{max}$ as the amplitude scales at which the breaking parameter turns into 1

$$Br = \frac{\omega^2 R_{max}}{\alpha^2 g} = 1, \qquad (5.1)$$

And to use dimensionless displacement ($y=r/R_{max}$). Then the dimensionless amplitude is

$$A = \frac{R_0}{R_{max}} \leq 1 , \qquad (5.2)$$

and formula (4.3) is converted to the form *(-A <y <A)*



$$W_y^{\sin}(y;A) = \frac{1}{\pi} \frac{1}{\sqrt{A^2 - \left[1 - \sqrt{1 + 2y + A^2}\right]^2}} \ . \tag{5.3}$$

Assuming now that the wave amplitude $A$ is a random variable, we average (5.3) by using
the amplitude distribution density $W_A(A)$
$$W(y) = \int_y^\infty W_y^{\sin}(y;A) \, \mathrm{W}_A(A) dA \ . \tag{5.4}$$

Formula (5.4) has an important practical meaning: by the measured distribution of the
wave amplitudes far from the coast (re-computed on run-up amplitudes in the linear theory), it is
possible to obtain the distribution of the wave run-up characteristics on the coast. The only
requirement imposed on the wave ensemble is that it should not contain breaking waves, which
should be somehow removed from the record. It immediately follows that the Gaussian field
containing large amplitude tails does not fit this requirement, and it should be modified.
Therefore, we assume the amplitude distribution to be finite for $A < A_{max} = 1$. In this case, the
breaking will not be implemented in any way, and the random wave run-up will take place
without any breaking. Further calculations depend on the specific type of the amplitude
distribution.
Let us construct the finite amplitude distribution at which the linear field distribution is
close to the Gaussian form and modify the Rayleigh distribution in the area $A < A_{max} = 1$ (Fig. 5)
$$W_A(A; A_{\max}, A_s) = \frac{1}{1 - \exp(-2A_{\max}^2 / A_s^2)} \frac{4A}{A_s^2} \exp\left(-2\frac{A^2}{A_s^2}\right), A \le A_{\max}, \tag{5.5}$$

to make the density function distribution normalized. Here, $A_s$ is the so-called significant wave
run-up height (an averaged value of 1/3 highest amplitudes). We would like to note here, that it
follows from (2.11) and (2.12) that the extremal run-up characteristics in the nonlinear theory
remain the same as in the linear theory. This means that the significant wave run-up height
remains the same as in the nonlinear theory.

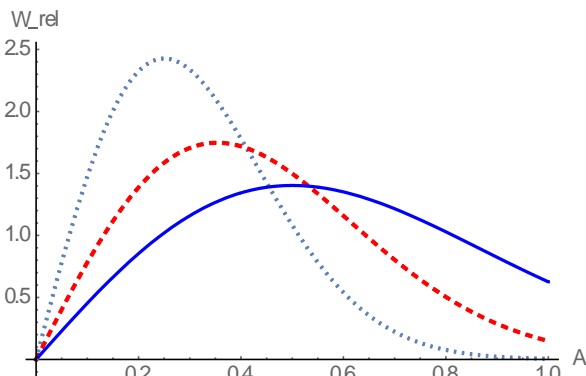



Fig. 5. The modified Rayleigh distribution (5.5) for different distribution values $A_s/A_{max}$;
0.5 –the dotted curve, 0.7 –the dashed line, 1 –the solid line.

When $A_s << A_{max} = 1$, distribution (5.5) transforms into the Rayleigh one, which is
characteristic of the Gaussian initial distribution of a narrow-band random signal. With the help
of (5.5), it becomes possible to calculate the distribution function of shoreline oscillations for the
various wave energy. So, with the incident wave small amplitude ($A_s << 1$), distribution (5.3) can
be replaced by a simpler expression (4.4) and the answer is the run-up distribution characteristics
in the linear theory:
$$W_{lin}(y; A_{max}, A_s) = \frac{4}{\pi A_s^2 [1 - \exp(-2A_{max}^2 / A_s^2)]} \int_y^{A_{max}} \frac{A}{\sqrt{A^2 - y^2}} \exp\left(-2\frac{A^2}{A_s^2}\right) dA. \qquad (5.6)$$

Besides, if $A_s << A_{max} = 1$, the integral (5.6) is reduced to the Gaussian distribution
$$W_{lin}(y; A_s) = \frac{2}{\sqrt{2\pi}A_s} \exp\left(-2\frac{y^2}{A_s^2}\right), \qquad (5.7)$$

where, $A_s = 2\sigma_y$, and $\sigma_y^2$ is the moving shoreline oscillation dispersion.
Fig. 6 shows the distribution of the run-up characteristics for different ratios of $A_s/A_{max}$
values by formulas (5.4) and (5.5); they are shown in solid lines. Here the dashed lines show the
calculation results according to the linear theory (5.6). As one can see, with $A_s/A_{max} = 0.5$ (the top
panel) and 0.7 (the middle panel), the linear distribution is close to the Gaussian one.
Nonlinearity leads to the asymmetry of the distribution function density in the direction of
positive values corresponding to the wave characteristics on the coast. If the undisturbed wave
ensemble is made of relatively large waves ($A_s/A_{max} = 1$), their distribution is far from the
Gaussian, both in the linear and in the nonlinear approximation.





Fig. 6. The probabilistic density function of the vertical shoreline displacement in the nonlinear theory (solid lines) and in the linear theory (dashed lines) for different $A_s/A_{max}$: 0.5 values: (the upper panel), 0.7 (the middle panel) and 1 (the lower panel).



The finite ($A<A_{max}$) power-law distribution concentrated mainly near the maximum

amplitude $A_{max}$ can be considered as another example of undisturbed large-amplitude waves.
$$W_A(A) = \frac{6A^5}{A_{max}^6}. \tag{5.8}$$

Fig. 7 shows the graphs of the probabilistic density function of the moving shoreline
displacement calculated by using formula (5.4) and (4.4) in the linear theory and (5.3) in the
nonlinear theory. It is also seen in the figure that nonlinear effects lead to a strong asymmetry
towards the positive values, that is, to the wave amplification at the run-up up stage than at the
run-down stage.

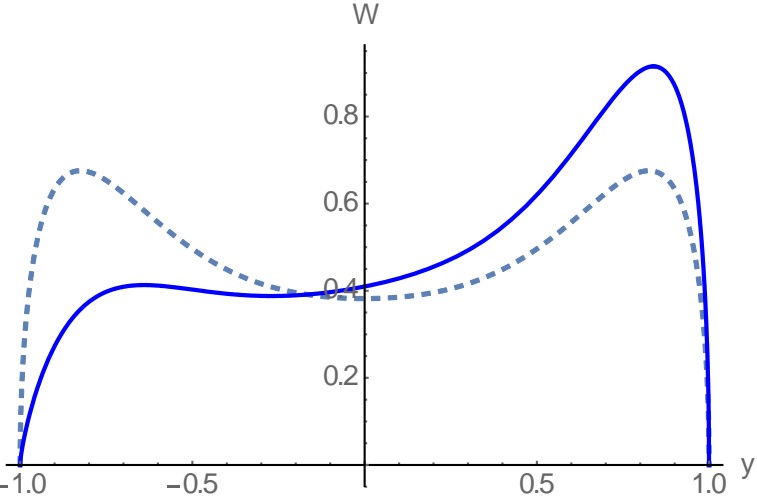


Fig. 7. Probabilistic density function of the shoreline vertical displacement in the linear

theory (dashed line) and non-linear theory (solid line)

**6. The wave breaking effect on probabilistic run-up characteristics**

The theory described above is valid for non-breaking waves. The mentioned wave ensemble,

strictly speaking, cannot be the Gaussian one, as it always has unlimited tails in the probability
density function. Let us briefly discuss what the formulas obtained for non-breaking waves lead
to in the presence of broken waves. Fig. 8 shows the parametric curve (3.11) - (3.12) when $Br =$
2. Formally, the curve became multi-valued in the range of negative values corresponding to the
maximum water outflow from the coast. We have already indicated that the probability density
function of the moving shoreline vertical displacement $W(\xi)$ coincides with the relative
residence time $\xi(t)$ of the function in the interval ($\xi$, $\xi+d\xi$), which is calculated by formula



(3.1). In contrast to negative cut-off bias values, in the area of positive values there is no
ambiguity, and, therefore, all the calculations can be carried out by using the formulas described
above. An example of such calculation with $Br = 2$ and $r > -0.5$ (in the zone of one-value
solution) is shown in Fig. 9.

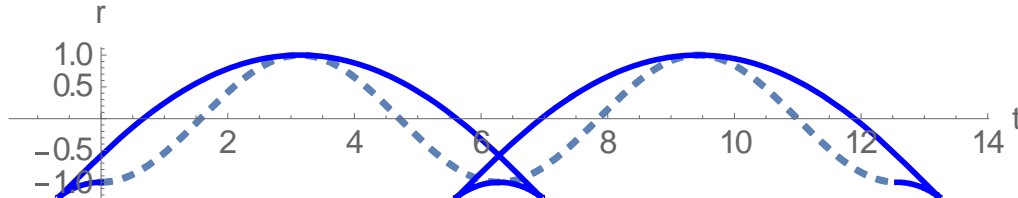



Fig. 8. The parametric curve (3.11) - (3.12) with $Br = 2$ (the solid curve) in comparison with the
linear problem with $Br = 0$ (the dashed line)

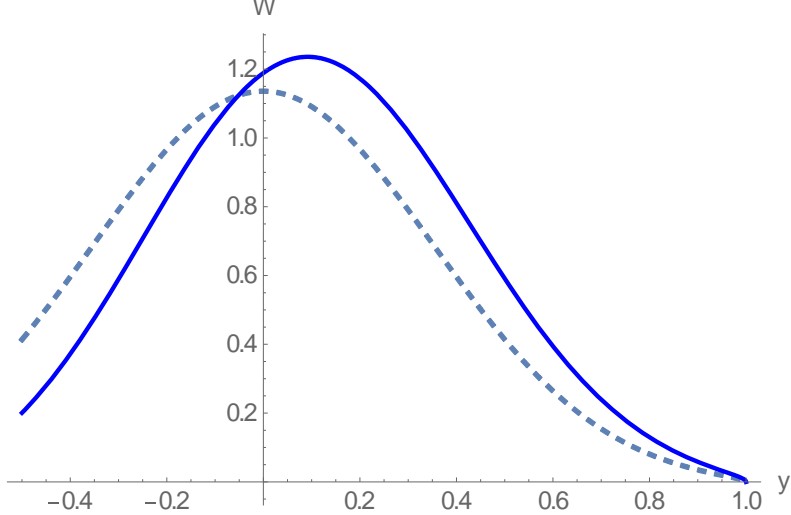


Fig. 9. The probability density function at $Br = 2$, constructed by formulas (5.3), (5.4) and (5.5)
(the solid line) in comparison with the linear distribution (5.6) is the dotted line. $A_s/A_{max} = 0.7$.

However, these results should be treated with caution, since after the wave breaking at the run-
down stage; it is not obvious that the next wave climbing on the coast will not break. This
important issue requires going beyond the theory discussed in this article.





## 7. Discussion and conclusion

In this paper, we study the run-up of irregular narrow-band waves with a random envelope (swell, storm surges, and tsunami) on a beach of a constant slope. The work was carried out in the framework of the nonlinear wave theory with one important assumption: there should be no breaking waves in the wave ensemble. This restriction is quite strict for field and laboratory conditions, but nevertheless, there are cases when it is performed. For instance, 75% of historical tsunami waves climbed on the coast with no breaking (Mazova et al, 1983). In the experiments performed in the Warwick University tank and in the Large Tank in Hannover (Denissenko et al, 2011, 2013), this condition was fulfilled.

The wave nonlinearity at the run-up stage leads to increased deviations from Gaussianity, as might be expected from general considerations. Nevertheless, it is shown that the probability distribution of the moving shoreline velocity does not depend on the wave nonlinearity and can be calculated within the linear theory framework. The same conclusion can be drawn about the distribution of the extreme run-up characteristics (the moving shoreline displacement and speed), which, in fact, has already been discussed earlier (Didenkulova et al, 2008). However, the probabilistic density function of the moving shoreline displacement differs from that predicted one in the linear theory framework. It is described by formula (5.4) by using either the theoretical or the measured distribution of the incident wave amplitudes. The paper gives the calculation results of the probable run-up characteristics with a modified Rayleigh distribution for wave amplitudes.

The wave breaking leads to the inapplicability of the wave run-up theory based on the Carrier-Greenspan transformation. If, nevertheless, the share of large amplitude waves is small, the breaking occurs mainly at the run-down stage, having little effect on the long-wave coast flooding characteristics (see Section 6). This question, however, requires a special study based on direct numerical solutions of the shallow-water equations or their nonlinear-dispersive generalizations.

Finally, it is worth noting that we considered the narrow-band wave run-up with a random amplitude and phase; as for the random waves with a wide spectrum – it is the problem of further consideration.

Obtained probability density functions of the vertical displacement of the moving shoreline are useful to compute statistical characteristics of flooding time and force on coasts and constructions, which are necessity for mitigation of natural marine hazards.



**Acknowledgment:**

The work is supported by the grants from the Russian Science Foundation: No.19-12-00256 (in part of computing the random Riemann wave characteristics) and No. 19-12-00253 (in part of computations the probability density function of the moving shoreline).

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
