# Peer review of "Probabilistic characteristics of narrow-band long wave run-up onshore"

_Natural Hazards and Earth System Sciences, 2019_

## Referee Comment (RC1) · Anonymous Referee #1 · 27 Jun 2019

The NHESS paper 2019-176 "Probabilistic characteristics of narrow-band long wave run-up onshore" by Gurbatov and Pelinovsky presents an interesting analysis of random, long-wave runup with amplitudes and phases of offshore waves defined probabilistically. The paper is well organized and, except for some minor clarifications listed below, is well written. Important conclusions are given with regard to the validity of linear theory for runup and inundation probability distributions. Given the scope of the journal, it would be advisable to indicate how the results from this study impact current probabilistic long-wave hazard assessments, as indicated in Comment 1. Overall, the nature of the comments below, in my opinion, are minor. Upon revision, this paper should be an important contribution to NHESS.

Technical comments:

[Figure]

(1) For probabilistic tsunami hazard assessments (PTHAs) in particular, there have been several recent studies that approximate runup and inundation from a probabilistic determination of offshore wave characteristics as summarized by Grezio et al. (2017). For example, Lorito et al. (2015) use a Green's Law approximation to estimate inundation. Davies et al. (2017) use an "amp-factor" method derived from Løvholt et al. (2012). Similarly, Mueller et al. (2015) use "linear predictors" to estimate runup. Can the results of the authors' study be used to evaluate these various PTHA runup/inundation estimators? (2) L36: Løvholt et al. (2012) indicate that the hydrostatic assumption reduces runup variability, compared to including dispersion. (3) L46-56: Should also probably summarize the work of Carrier (1995) and Carrier et al. (2003). (4) L111-112: It is worth noting that Carrier (1995) also derives runup from along-shore (i.e., edge wave) propagation. (5) Eqns. 2.5-2.8: Carrier (1995) includes quadratic terms in these equations, deemed negligible. (6) It might not be advisable to include Section 6, since as the authors indicate, the complex interaction of breaking waves is not included.

Grammatical/typographical comments

(7) Citation formatting: when the authors are part of the sentence, do not place in parentheses (L46, 64, 171-173, 186). (8) L29-31: Important first sentence is awkwardly constructed. (9) L41: Space between "linearized" and "by". (10) L58-59: "Moreover, very often the leading wave turns out <not> to be the maximum one." (11) L62: "their help" is confusing. (12) L80 and elsewhere: Most likely "simple" wave equation will be misunderstood by most readers as an alternative name for the Riemann wave equation. (13) L99: "climbs"->"approaches" (14) L169: Which equation does "ODE" refer to? (15) L182: Remove hyphen before Br (could be interpreted as a negative sign) (16) L184: What does "last sea particle acceleration" mean? (17) L224-225: Awkward sentence. (18) L238: "what is another record" -> "which is another expression" (19) Fig. 2 caption: Indicate that this is for monochromatic waves? (20) L266-267: insert "W" after "vertical displacement" (correct?) How is W related to R, as a random variate? (21) L274:

Replace Russian character for "and" with English equivalent. (22) L308: "the equation mentioned last" -> "the last equation" (23) L351: Indicate that the Rayleigh distribution is for wave heights.

References

Carrier, G.F., 1995. On-shelf tsunami generation and coastal propagation. in Tsunami: Progress in Prediction, Disaster Prevention and Warning, pp. 1-20, eds. Tsuchiya, Y. & Shuto, N. Kluwer, Dordrecht, The Netherlands.

Carrier, G.F., Wu, T.T. & Yeh, H., 2003. Tsunami run-up and draw-down on a plane beach, Journal of Fluid Mechanics, 475, 79-99.

Davies, G., Griffin, J., Løvholt, F., Glimsdal, S., Harbitz, C., Thio, H.K., Lorito, S., Basili, R., Selva, J., Geist, E.L. & Baptista, M.A., 2017. A global probabilistic tsunami hazard assessment from earthquake sources. in Tsunamis: Geology, Hazards and Risks, pp. doi:10.1144/SP1456.1146, eds. Scourse, E. M., Chapman, N. A., Tappin, D. R. & Wallis, S. R. Geological Society of London Spec. Pub. 456, London.

Grezio, A., Babeyko, A.Y., Baptista, A.M., Behrens, J., Costa, A., Davies, G., Geist, E.L., Glimsdal, S., González, F.I., Griffin, J., Harbitz, C.B., LeVeque, R.J., Lorito, S., Løvholt, F., Omira, R., Mueller, C.S., Paris, R., Parsons, T., Polet, J., Power, W., Selva, J., Sørensen, M.B. & Thio, H.K., 2017. Probabilistic Tsunami Hazard Analysis (PTHA): Multiple sources and global applications, Reviews of Geophysics, 55, 1158-1198.

Lorito, S., Selva, J., Basili, R., Romano, F., Tiberti, M.M. & Piatanesi, A., 2015. Probabilistic hazard for seismically induced tsunamis: accuracy and feasibility of inundation maps, Geophys. J. Int., 200, 574-588.

Løvholt, F., Pedersen, G., Bazin, S., Kühn, D., Bredesen, R.E. & Harbitz, C., 2012. Stochastic analysis of tsunami runup due to heterogeneous coseismic slip and dispersion, J. Geophys. Res., 117, doi:10.1029/2011JC007616.

Mueller, C., Power, W., Fraser, S. & Wang, X., 2015. Effects of rupture complexity on

local tsunami inundation: Implications for probabilistic tsunami hazard assessment by example, Journal of Geophysical Research: Solid Earth, 120, 488-502.

---

## Referee Comment (RC2) · Anonymous Referee #2 · 3 Jul 2019

The paper presents a theoretical study of random long wave run-up over a plane beach. It starts with a general introduction of the well-known Carrier-Greenspan approach and then describes linear and nonlinear shoreline dynamics of monochromatic waves. The early sections provide reviews of previous works by the authors. The novelty of this work lies in the probabilistic analysis of shoreline displacement and velocity in the latter sections. The authors apply the geometric probability theory for shoreline dynamics to compare statistical properties of linear and nonlinear wave run-up on the shore. Although the approach has significant limitations (e.g. non-breaking and non-dispersive long waves), the paper provides a statistical view of nonlinear wave run-up which is of interest to the community. I recommend publication of the paper after following comments are addressed.

[Figure]

-There are typos, missing spaces between words and grammatical errors. Please edit the paper carefully.

-The equation (4.1) is a bit confusing. The RHS of the equation appears to have dimension after reading from the previous sections. Please improve the notation for readers who are not very familiar with the geometric probability theory.

-The assumption of "narrow band" is not clearly explained. In section 5, the incident wave is introduced as "a quasi-harmonic wave with a random amplitude and phase" (L328). The authors do not mention anything about wave period.

-L340-349: Is it obvious that narrow-band random waves exhibit non-breaking wave run-up if individual monochromatic waves are below the breaking criterion? This seems to require certain assumptions or some explanation at least.

-The result of broken wave runup in Section 6 may be questionable. The setting with Br=2 implies that wave breaking occurs before the incident waves arrive at the shore (The Jacobian breaks down seawards of the shoreline). This may affect the probabilistic distribution by eliminating the tail on the positive side.

---

## Author Comment (AC1) · 22 Jul 2019

**Probabilistic characteristics of narrow-band long wave run-up onshore**
**by Sergey Gurbatov and Efim Pelinovsky**

First of all, we would like to thank two anonymous reviewers for their useful comments and suggestions. An item-by-item response on all the comments is presented below.

**Referee #1**

*The NHESS paper 2019-176 "Probabilistic characteristics of narrow-band long wave run-up onshore" by Gurbatov and Pelinovsky presents an interesting analysis of random, long-wave runup with amplitudes and phases of offshore waves defined probabilistically. The paper is well organized and, except for some minor clarifications listed below, is well written. Important conclusions are given with regard to the validity of linear theory for runup and inundation probability distributions. Given the scope of the journal, it would be advisable to indicate how the results from this study impact current probabilistic long-wave hazard assessments, as indicated in Comment 1. Overall, the nature of the comments below, in my opinion, are minor. Upon revision, this paper should be an important contribution to NHESS.*

*Technical comments:*

*(1) For probabilistic tsunami hazard assessments (PTHAs) in particular, there have been several recent studies that approximate runup and inundation from a probabilistic determination of offshore wave characteristics as summarized by Grezio et al. (2017). For example, Lorito et al. (2015) use a Green's Law approximation to estimate inundation. Davies et al. (2017) use an "amp-factor" method derived from Løvholt et al. (2012). Similarly, Mueller et al. (2015) use "linear predictors" to estimate runup. Can the results of the authors' study be used to evaluate these various PTHA runup/inundation estimators?*

Actually, it is a very important discussion connected with the applicability of various runup formulas. Some of them (for example, Green's Law) are particular cases of analytical formulas used in our approach. In fact, they are used to analyze the tsunami waves which are not a stationary random process. The study of such processes is beyond scope of our paper where we try to get analytical results in the case of input signals presented as the stationary random processes (swell, seishes, the atmospheric origin tsunami etc). We would not discuss in our paper this important discussion suggested by the reviewer.

*(2) L36: Løvholt et al. (2012) indicate that the hydrostatic assumption reduces runup variability, compared to including dispersion.*

It is an important comment, therefore, we added the final paragraph in conclusion: Now in practice various generalizations of shallow-water equations are used to analyse the tsunami runup including wave dispersion, see, for instance (Lovholt et al, 2012). Wave dispersion as a quadratic dissipative term prevents us from getting analytical results, so their influence on statistical characteristics should be investigated in future.

*(3) L46-56: Should also probably summarize the work of Carrier (1995) and Carrier et al. (2003).*

These papers are included in the list of references.

*(4) L111-112: It is worth noting that Carrier (1995) also derives runup from along-shore (i.e., edge wave) propagation.*

Yes, we know these results were also published in the JFM paper as well as the results given by Brocchini. However, these results are appoximated and not quite good for the rigorous theory.

*(5) Eqns. 2.5-2.8: Carrier (1995) includes quadratic terms in these equations, deemed negligible.*

The quadratic dissipative term is widely used in practice, but in the rigorous benchmark theory there are no analytical results, and the analysis of such equations are beyond scope of this paper.

*(6) It might not be advisable to include Section 6, since as the authors indicate, the complex interaction of breaking waves is not included.*

We absolutely agree with this comment. That is why our text after Fig. 9 contains the following conclusion: "This important issue requires going beyond the theory discussed in this article". We slightly modified the final paragraph going after Fig. 9 by saying:
However, these results should be treated with caution. If $Br > 1$ the Jacobian breaks down seawards of the shoreline. This may affect the probabilistic distribution on the positive side. This important issue requires going beyond the theory discussed in this article

Grammatical/typographical comments:
(7) Citation formatting: when the authors are part of the sentence, do not place in parentheses (L46, 64, 171-173, 186).
Done

*(8) L29-31: Important first sentence is awkwardly constructed.*

The sentence has been modified and runs as follows: The flooded area size, the water flow depth and its speed on the coast, the coastal topography characteristics determine the consequences of marine natural disasters on the coast

*(9) L44: Space between "linearized" and "by".*

Done

*(10) L58-59: "Moreover, very often the leading wave turns out <not> to be the maximum one."*

The sentence is modified: Moreover, very often the leading wave is not the maximum one.

*(11) L62: "their help" is confusing.*

Deleted

*(12) L80 and elsewhere: Most likely "simple" wave equation will be misunderstood by most readers as an alternative name for the Riemann wave equation.*

Unfortunately, the term "the simple wave equation" is used more often than "the Riemann wave equation". It is why we would like to use both terms.

(13) L99: "climbs"->"approaches"

Done

*(14) L169: Which equation does "ODE" refer to?*

Corrected, the following items have been inserted:  Eqs. (2.11) and (2.12)

*(15) L182: Remove hyphen before Br (could be interpreted as a negative sign)*

Done

*(16) L184: What does "last sea particle acceleration" mean?*

The last sea particle acceleration ($\alpha^{-1}d^2R/dt^2$) means the acceleration of the moving shoreline along the slope in the linear theory.

*(17) L224-225: Awkward sentence.*

The sentence "Formula (3.6) allows working further with the run-up height $R_0$ instead of the wave amplitude far from the coast $a(x)$, considering it to be given" replaced by:
Formula (3.6) allows working further with the run-up height $R_0$ instead of the wave amplitude far from the coast $a(x)$. This run-up height will be considered as the given value.

*(18) L238: "what is another record" -> "which is another expression"*

Done

*(19) Fig. 2 caption: Indicate that this is for monochromatic waves?*

Added: in the case of the incident monochromatic wave

*(20) L266-267: insert "W" after "vertical displacement" (correct?) How is W related to R, as a random variate?*

Thank you for the comment, Eq. (4.1) is now re-written in the dimensionless form, and all the values are understood. In fact, $W(z)dz=W(r)dr$, and, therefore, $W(r)=W(z=r/R_0)/R_0$

*(21) L274: Replace Russian character for "and" with English equivalent.*

Done

*(22) L308: "the equation mentioned last" -> "the last equation"*

Changed into equation (3.12)

*(23) L351: Indicate that the Rayleigh distribution is for wave heights.*

Corrected. It is now given in L353.

**Referee #2**

*The paper presents a theoretical study of random long wave run-up over a plane beach. It starts with a general introduction of the well-known Carrier-Greenspan approach and then describes linear and nonlinear shoreline dynamics of monochromatic waves. The early sections provide reviews of previous works by the authors. The novelty of this work lies in the probabilistic analysis of shoreline displacement and velocity in the latter sections. The authors apply the geometric probability theory for shoreline dynamics to compare statistical properties of linear and nonlinear wave run-up on the shore. Although the approach has significant limitations (e.g. non-breaking and nondispersive long waves), the paper provides a statistical view of nonlinear wave runup which is of interest to the community. I recommend publication of the paper after following comments are addressed.*

*-There are typos, missing spaces between words and grammatical errors. Please edit the paper carefully.*

Corrected

*-The equation (4.1) is a bit confusing. The RHS of the equation appears to have dimension after reading from the previous sections. Please improve the notation for readers who are not very familiar with the geometric probability theory.*

Thank you for the comment. We have re-written equation (4.1) in dimensionless variables. This comment is also used to modify Fig. 2 in the dimensionless form.

*-The assumption of "narrow band" is not clearly explained. In section 5, the incident wave is introduced as "a quasi-harmonic wave with a random amplitude and phase" (L328). The authors do not mention anything about wave period.*

We have added the definition of the narrow-band wave field (see answer on next comments):
The narrow-band random wave field contains sine waves with almost constant frequency and random amplitude and phase.

*-L340-349: Is it obvious that narrow-band random waves exhibit non-breaking wave run-up if individual monochromatic waves are below the breaking criterion? This seems to require certain assumptions or some explanation at least.*

We slightly modified the text in lines L340-349:

Formula (5.4) has an important practical meaning: by the measured distribution of the wave amplitudes far from the coast (re-computed on run-up amplitudes in the linear theory), it is possible to obtain the distribution of the wave run-up characteristics on the coast. The only requirement imposed on the wave ensemble is that it should not contain breaking waves, which should be somehow removed from the record. It immediately follows that the Gaussian field containing large amplitude tails does not fit this requirement, and it should be modified. Therefore, we assume the amplitude distribution to be finite for $A < A_{max} = 1$. The narrow-band random wave field contains sine waves with almost constant frequency and random amplitude and phase. It means that if the wave amplitude is below the "breaking amplitude" $A_{max} = 1$, the breaking will not be implemented in any way, and the random wave run-up will take place without any breaking. Further calculations depend on the specific type of the amplitude distribution.

*-The result of broken wave runup in Section 6 may be questionable. The setting with Br=2 implies that wave breaking occurs before the incident waves arrive at the shore (The Jacobian breaks down seawards of the shoreline). This may affect the probabilistic distribution by eliminating the tail on the positive side.*

We absolutely agree with this comment. That is why our text after Fig. 9 contains the following conclusion: "
[revised manuscript text omitted]

---

## Author Comment (AC3) · 2 Aug 2019

**Probabilistic characteristics of narrow-band long wave run-up onshore**
**by Sergey Gurbatov and Efim Pelinovsky**

First of all, we would like to thank two anonymous reviewers for their useful comments and suggestions. An item-by-item response on all the comments is presented below.

**Referee #1**

*The NHESS paper 2019-176 "Probabilistic characteristics of narrow-band long wave run-up onshore" by Gurbatov and Pelinovsky presents an interesting analysis of random, long-wave runup with amplitudes and phases of offshore waves defined probabilistically. The paper is well organized and, except for some minor clarifications listed below, is well written. Important conclusions are given with regard to the validity of linear theory for runup and inundation probability distributions. Given the scope of the journal, it would be advisable to indicate how the results from this study impact current probabilistic long-wave hazard assessments, as indicated in Comment 1. Overall, the nature of the comments below, in my opinion, are minor. Upon revision, this paper should be an important contribution to NHESS.*

*Technical comments:*

*(1) For probabilistic tsunami hazard assessments (PTHAs) in particular, there have been several recent studies that approximate runup and inundation from a probabilistic determination of offshore wave characteristics as summarized by Grezio et al. (2017). For example, Lorito et al. (2015) use a Green's Law approximation to estimate inundation. Davies et al. (2017) use an "amp-factor" method derived from Løvholt et al. (2012). Similarly, Mueller et al. (2015) use "linear predictors" to estimate runup. Can the results of the authors' study be used to evaluate these various PTHA runup/inundation estimators?*

Actually, it is a very important discussion connected with the applicability of various runup formulas. Some of them (for example, Green's Law) are particular cases of analytical formulas used in our approach. In fact, they are used to analyze the tsunami waves which are not a stationary random process. The study of such processes is beyond scope of our paper where we try to get analytical results in the case of input signals presented as the stationary random processes (swell, seishes, the atmospheric origin tsunami etc). We would not discuss in our paper this important discussion suggested by the reviewer.

*(2) L36: Løvholt et al. (2012) indicate that the hydrostatic assumption reduces runup variability, compared to including dispersion.*

It is an important comment, therefore, we added the final paragraph in conclusion: Now in practice various generalizations of shallow-water equations are used to analyse the tsunami runup including wave dispersion, see, for instance (Lovholt et al, 2012). Wave dispersion as a quadratic dissipative term prevents us from getting analytical results, so their influence on statistical characteristics should be investigated in future.

*(3) L46-56: Should also probably summarize the work of Carrier (1995) and Carrier et al. (2003).*

These papers are included in the list of references.

*(4) L111-112: It is worth noting that Carrier (1995) also derives runup from along-shore (i.e., edge wave) propagation.*

Yes, we know these results were also published in the JFM paper as well as the results given by Brocchini. However, these results are appoximated and not quite good for the rigorous theory.

*(5) Eqns. 2.5-2.8: Carrier (1995) includes quadratic terms in these equations, deemed negligible.*

The quadratic dissipative term is widely used in practice, but in the rigorous benchmark theory there are no analytical results, and the analysis of such equations are beyond scope of this paper.

*(6) It might not be advisable to include Section 6, since as the authors indicate, the complex interaction of breaking waves is not included.*

We absolutely agree with this comment. That is why our text after Fig. 9 contains the following conclusion: "This important issue requires going beyond the theory discussed in this article". We slightly modified the final paragraph going after Fig. 9 by saying:
However, these results should be treated with caution. If $Br > 1$ the Jacobian breaks down seawards of the shoreline. This may affect the probabilistic distribution on the positive side. This important issue requires going beyond the theory discussed in this article

Grammatical/typographical comments:
(7) Citation formatting: when the authors are part of the sentence, do not place in parentheses (L46, 64, 171-173, 186).
Done

*(8) L29-31: Important first sentence is awkwardly constructed.*

The sentence has been modified and runs as follows: The flooded area size, the water flow depth and its speed on the coast, the coastal topography characteristics determine the consequences of marine natural disasters on the coast

*(9) L44: Space between "linearized" and "by".*

Done

*(10) L58-59: "Moreover, very often the leading wave turns out <not> to be the maximum one."*

The sentence is modified: Moreover, very often the leading wave is not the maximum one.

*(11) L62: "their help" is confusing.*

Deleted

*(12) L80 and elsewhere: Most likely "simple" wave equation will be misunderstood by most readers as an alternative name for the Riemann wave equation.*

Unfortunately, the term "the simple wave equation" is used more often than "the Riemann wave equation". It is why we would like to use both terms.

(13) L99: "climbs"->"approaches"

Done

*(14) L169: Which equation does "ODE" refer to?*

Corrected, the following items have been inserted:  Eqs. (2.11) and (2.12)

*(15) L182: Remove hyphen before Br (could be interpreted as a negative sign)*

Done

*(16) L184: What does "last sea particle acceleration" mean?*

The last sea particle acceleration ($\alpha^{-1} d^2 R / dt^2$) means the acceleration of the moving shoreline along the slope in the linear theory.

*(17) L224-225: Awkward sentence.*

The sentence "Formula (3.6) allows working further with the run-up height $R_0$ instead of the wave amplitude far from the coast $a(x)$, considering it to be given" replaced by:
Formula (3.6) allows working further with the run-up height $R_0$ instead of the wave amplitude far from the coast $a(x)$. This run-up height will be considered as the given value.

*(18) L238: "what is another record" -> "which is another expression"*

Done

*(19) Fig. 2 caption: Indicate that this is for monochromatic waves?*

Added: in the case of the incident monochromatic wave

*(20) L266-267: insert "W" after "vertical displacement" (correct?) How is W related to R, as a random variate?*

Thank you for the comment, Eq. (4.1) is now re-written in the dimensionless form, and all the values are understood. In fact, $W(z)dz=W(r)dr$, and, therefore, $W(r)=W(z=r/R_0)/R_0$

*(21) L274: Replace Russian character for "and" with English equivalent.*

Done

*(22) L308: "the equation mentioned last" -> "the last equation"*

Changed into equation (3.12)

*(23) L351: Indicate that the Rayleigh distribution is for wave heights.*

Corrected. It is now given in L353.

***References:***
*Carrier, G.F., 1995. On-shelf tsunami generation and coastal propagation. in Tsunami: Progress in Prediction, Disaster Prevention and Warning, pp. 1-20, eds. Tsuchiya, Y. & Shuto, N. Kluwer, Dordrecht, The Netherlands.*
*Carrier, G.F., Wu, T.T. & Yeh, H., 2003. Tsunami run-up and draw-down on a plane beach, Journal of Fluid Mechanics, 475, 79-99.*
*Davies, G., Griffin, J., Løvholt, F., Glimsdal, S., Harbitz, C., Thio, H.K., Lorito, S., Basili, R., Selva, J., Geist, E.L. & Baptista, M.A., 2017. A global probabilistic tsunami hazard assessment from earthquake sources. in Tsunamis: Geology, Hazards and Risks, pp. doi:10.1144/SP1456.1146, eds. Scourse, E. M., Chapman, N. A., Tappin, D. R. & Wallis, S. R. Geological Society of London Spec. Pub. 456, London.*
*Grezio, A., Babeyko, A.Y., Baptista, A.M., Behrens, J., Costa, A., Davies, G., Geist, E.L., Glimsdal, S., González, F.I., Griffin, J., Harbitz, C.B., LeVeque, R.J., Lorito, S., Løvholt, F., Omira, R., Mueller, C.S., Paris, R., Parsons, T., Polet, J., Power, W., Selva, J., Sørensen, M.B. & Thio, H.K., 2017. Probabilistic Tsunami Hazard Analysis (PTHA): Multiple sources and global applications, Reviews of Geophysics, 55, 1158-1198.*
*Lorito, S., Selva, J., Basili, R., Romano, F., Tiberti, M.M. & Piatanesi, A., 2015. Probabilistic hazard for seismically induced tsunamis: accuracy and feasibility of inundation maps, Geophys. J. Int., 200, 574-588.*
*Løvholt, F., Pedersen, G., Bazin, S., Kühn, D., Bredesen, R.E. & Harbitz, C., 2012. Stochastic analysis of tsunami runup due to heterogeneous coseismic slip and dispersion, J. Geophys. Res., 117, doi: 10.1029/2011JC007616.*
*Mueller, C., Power, W., Fraser, S. & Wang, X., 2015. Effects of rupture complexity on local tsunami inundation: Implications for probabilistic tsunami hazard assessment by example, Journal of Geophysical Research: Solid Earth, 120, 488-502.*

**Referee #2**

*The paper presents a theoretical study of random long wave run-up over a plane beach. It starts with a general introduction of the well-known Carrier-Greenspan approach and then describes linear and nonlinear shoreline dynamics of monochromatic waves. The early sections provide reviews of previous works by the authors. The novelty of this work lies in the probabilistic analysis of shoreline displacement and velocity in the latter sections. The authors apply the geometric probability theory for shoreline dynamics to compare statistical properties of linear and nonlinear wave run-up on the shore. Although the approach has significant limitations (e.g. non-breaking and nondispersive long waves), the paper provides a statistical view of nonlinear wave runup which is of interest to the community. I recommend publication of the paper after following comments are addressed.*

*-There are typos, missing spaces between words and grammatical errors. Please edit the paper carefully.*

Corrected

*-The equation (4.1) is a bit confusing. The RHS of the equation appears to have dimension after reading from the previous sections. Please improve the notation for readers who are not very familiar with the geometric probability theory.*

Thank you for the comment. We have re-written equation (4.1) in dimensionless variables. This comment is also used to modify Fig. 2 in the dimensionless form.

*-The assumption of "narrow band" is not clearly explained. In section 5, the incident wave is introduced as "a quasi-harmonic wave with a random amplitude and phase" (L328). The authors do not mention anything about wave period.*

We have added the definition of the narrow-band wave field (see answer on next comments):
The narrow-band random wave field contains sine waves with almost constant frequency and random amplitude and phase.

*-L340-349: Is it obvious that narrow-band random waves exhibit non-breaking wave run-up if individual monochromatic waves are below the breaking criterion? This seems to require certain assumptions or some explanation at least.*

We slightly modified the text in lines L340-349:

Formula (5.4) has an important practical meaning: by the measured distribution of the wave amplitudes far from the coast (re-computed on run-up amplitudes in the linear theory), it is possible to obtain the distribution of the wave run-up characteristics on the coast. The only requirement imposed on the wave ensemble is that it should not contain breaking waves, which should be somehow removed from the record. It immediately follows that the Gaussian field containing large amplitude tails does not fit this requirement, and it should be modified. Therefore, we assume the amplitude distribution to be finite for $A < A_{max} = 1$. The narrow-band random wave field contains sine waves with almost constant frequency and random amplitude and phase. It means that if the wave amplitude is below the "breaking amplitude" $A_{max} = 1$, the breaking will not be implemented in any way, and the random wave run-up will take place without any breaking. Further calculations depend on the specific type of the amplitude distribution.

*-The result of broken wave runup in Section 6 may be questionable. The setting with Br=2 implies that wave breaking occurs before the incident waves arrive at the shore (The Jacobian breaks down seawards of the shoreline). This may affect the probabilistic distribution by eliminating the tail on the positive side.*

We absolutely agree with this comment. That is why our text after Fig. 9 contains the following conclusion: "This important issue requires going beyond the theory discussed in this article". We slightly modified the final paragraph that goes after Fig. 9 by saying:
However, these results should be treated with caution. If $Br > 1$ the Jacobian breaks down seawards of the shoreline. This may affect the probabilistic distribution on the positive side. This important issue requires going beyond the theory discussed in this article

---

## Author Response (AR2)

Author's response:

**Probabilistic characteristics of narrow-band long wave run-up onshore**

**Sergey Gurbatov and Efim Pelinovsky**

We thank Editor, Ira Didenkulova for all comments. Below is our answer.

*1) Runup or run-up? Please, select one and use it through out the paper.*

We use now only run-up.

*2) Figures. For all figures, ensure that the minimum font size is legible in your figures.*

We checked Figures and modified Fig 5.

*3) Spelling, grammar, sentence structure. Please, check again your manuscript. I made some corrections in Abstract and Introduction (see attached file).*

We checked manuscript and corrected.